# Measuring the Synergistic Effect of Pollution and Carbon Reduction in China's Industrial Sector

Minglong Xu [1], Huimin Li [1,*] and Xianghui Deng [2]

1   Beijing Climate Change Response Research and Education Center, Beijing University of Civil Engineering and Architecture, Beijing 100044, China; 2108570021019@stu.bucea.edu.cn
2   China Industrial Energy Conservation and Cleaner Production Association, Beijing 100034, China; dxh0819@126.com
*   Correspondence: lihm@bucea.edu.cn

**Abstract:** The industrial sector is a major source of $CO_2$ and atmospheric pollutants in China, and it is important to promote industrial pollution reduction and carbon reduction to improve the quality of China's atmospheric environment and meet $CO_2$ peak targets. In this paper, based on 2005 to 2021's panel data from the industrial sector, we construct a computational model of the synergistic effect of pollution reduction and carbon reduction, quantitatively evaluate the synergistic effect of industrial $CO_2$ emissions and air pollutants, and explore its evolutionary mechanism. The results showed that between 2005 and 2021, there was a clear synergistic effect between $CO_2$ and air pollutants in China's industrial sector, and the synergistic effect is increasing. For different pollutants, $CO_2$ and $SO_2$ have the strongest synergies, and $CO_2$ and particulate matter have relatively weak synergies. For different energy types, the synergies between coal-related carbon emissions and air pollutants gradually increase, while gas-related carbon emissions and pollutants tend to decrease. From different industry types, the synergies between $CO_2$ and air pollutants are weaker in high-polluting and high-emission industries than in other industries. These results have strong policy implications. First, the focus of synergistic measures should be on source reduction. The second is to make high-polluting and high-emission industries the focus of pollution reduction and carbon reduction. Third is harmonized management of air quality standards and carbon peaking should be promoted. The formulation of relevant policies from the above three aspects will help synergize pollution reduction and carbon reduction in the industrial sector.

**Keywords:** pollution and carbon reduction; synergistic effect; industrial sector; China

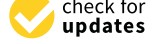



## 1. Introduction

The industrial sector in China represents the primary energy consumer, carbon emission contributor, and air pollutant source [1]. The industrial sector accounted for 68.6% of the country's total carbon emissions and 58.03% of pollutant emissions [2]. The Chinese government has recently prioritized the synergistic reduction of pollution and carbon as a fundamental policy, while emphasizing it as a crucial approach towards sustainable development [3]. In August 2015, the 'Law of the People's Republic of China on the Prevention and Control of Atmospheric Pollution' proposed for the first time the goal of "synergistic control of air pollutants and greenhouse gases" [4]. In 2016, the 'Work Program for Controlling Greenhouse Gas Emissions in the 13th Five-Year Plan' and '13th Five-Year Plan for Ecological and Environmental Protection' explicitly identified "strengthening the synergistic control of air pollutant emissions and carbon emissions" as a crucial means of achieving low-carbon transformation [5]. The Ministry of Ecology and Environment has explicitly emphasized the achievement of a synergistic effect between pollution reduction and carbon reduction as the target requirement for an in-depth battle against pollution prevention and control in 2021. Furthermore, promoting the coordinated governance of

pollution reduction and carbon reduction is regarded as a crucial approach to driving an overall green and low-carbon transformation in economic and social development.

Synergistic effects originated from the concomitant benefits proposed by Ayers et al. [6]. The IPCC formally introduced the concept of "synergistic effects" in its Third Assessment Report [7]. The consensus among scholars is that the concept of the "synergistic effect" encompasses the following several aspects: the policy aimed at reducing air-polluting emissions results in a synergistic effect of reducing both air pollutants and greenhouse gas emissions, while the policy targeting greenhouse gas emission reduction leads to a similar synergistic effect [8]. Additionally, the synergistic reduction of air pollutants and greenhouse gases resulting from coordinated regional management [9].

The existing literature predominantly concentrates on the synergistic impacts of pollution reduction and carbon mitigation in the subsequent domains: first, through verifying the existence of the synergistic effect between air pollutants and greenhouse gases. The possibility of synergistic emission reduction was confirmed by Wang et al. through an investigation into the distribution characteristics of atmospheric pollutants and greenhouse gas emissions, and the findings revealed that $SO_2$, particulate matter, and carbon exhibit similar emission trajectories [10]. The results of the study by Fu et al. show that $CO_2$-reduction activities in the power sector significantly affect the reduction of $SO_2$, confirming the possibility of the synergistic reduction of $CO_2$ and air pollutants in the power sector [11]. The synergistic effects of air pollutants and greenhouse gases exhibit regional variations, with the eastern and southern regions of China demonstrating more pronounced synergistic effects compared to the northern region [12]. The synergistic effect between $CO_2$ and particulate matter is more pronounced in the Beijing–Tianjin–Hebei region and the northern Yangtze River Delta region [13].

The second domain is to investigate the factors that influence the synergistic effects between atmospheric pollutants and greenhouse gases [14–20]. For example, Qian et al. [21] analyzed the potential synergistic effects of $SO_2$, $NO_X$, particulate matter, and $CO_2$ emissions from major industrial sectors in China by means of scenario simulation, and the results of the study showed that the improvement of energy intensity, the structural adjustment of the industrial scale, and the adjustment of the energy consumption structure can promote the synergistic emission reduction of air pollutants and $CO_2$.

The third domain involves assessing the combined impacts of policy implementation. Gao [22] and He et al. [23] evaluated the synergistic effects of energy saving and emission-reduction measures in the iron and steel industry and cement industry, respectively. The results show that measures in the "structural adjustment" category have a better synergistic-emission-reduction effect, but their abatement costs are higher than those in the "energy substitution" category, while measures in the "end-of-pipe" category are non-synergistic abatement measures with higher abatement costs due to the increase in energy consumption of the treatment equipment, which increases the emission of a certain amount of air pollutants or $CO_2$. Shi et al. [24] quantified the impact of China's clean air actions on energy use and $CO_2$ emissions from 2013 to 2020. The study showed that the implementation of measures such as upgrading industrial boilers, phasing out small and polluting enterprises, phasing out outdated industrial capacity, phasing out outdated capacity in the residential sector, and phasing out yellow-licensed vehicles avoided 57 million tons of $CO_2$ emissions. These measures are effective in promoting the synergistic emission reduction of air pollutants and $CO_2$. Meanwhile, the synergistic emission reduction from clean air actions far exceed the additional CO2 emissions from end-of-pipe facilities. The 'Environmental Protection Tax Law' [25], low-carbon city pilot [26], carbon pricing policy [27], clean energy policies [28], energy structure optimization [29], and electric vehicle promotion [30] all have synergistic-emission-reduction effects, significantly increasing the degree of synergy between air pollutants and $CO_2$. Therefore, the implementation of relevant emission-reduction policies is the fundamental reason for the synergistic effect between air pollutants and $CO_2$ [31].

The fourth domain involves projecting future trends in synergistic emission reductions. Based on the historical data of air pollutants and carbon emissions, future air pollutants as well as carbon emissions are predicted under different policy scenarios using models such as STIRPAT [32–35], LEAP [36,37] and CGE [38].

The existing studies have established crucial groundwork for comprehending the synergistic effect of pollution and carbon reduction, as well as exploring the path towards synergistic emission reduction. Nevertheless, the current research still possesses certain limitations as follows: assessing the synergistic impacts of individual policies but disregarding the interdependence among policy measures and the cumulative nature of policy effects [39–41]; analyzing the reasons for changes in synergistic effects in terms of the policies themselves but ignoring the underlying drivers of the policy measures [42,43]; focusing on synergy predictions for a single industry or a single city but failing to provide overall guidance on pollution and carbon reduction [44].

Combining the shortcomings of existing studies, this study provides a fresh perspective to understand the combined impact of pollution control and carbon emission reduction in the industrial sector. By examining the historical data on pollutant and carbon dioxide emissions, we identify synergistic features of China's industrial sector in reducing pollution and carbon emissions, while exploring feasible paths to enhance such synergies. The study develops a cross elasticity index to quantify the synergistic effects and calculates the industry-specific synergistic index from 2005 to 2021. To provide a comprehensive understanding of these effects, the sources of $CO_2$ are classified into distinct fuel categories, while pollutants are categorized as $SO_2$, $NO_X$, and particulate matter.

## 2. Methodology

### 2.1. Methods

During the period of 2005–2021, carbon emissions related to energy consumption in the industrial sector have surged from 3454.33 million tons to 7869.94 million tons, reflecting a remarkable growth rate of 128%. In contrast, air pollutant emissions have exhibited an overall declining trend, plummeting from 75.85 million tons to merely 11.63 million tons, indicating a significant reduction rate of 84.7%. From a total-amount perspective, no discernible synergistic relationship between air pollutants and carbon emissions can be observed. However, it is noteworthy that China's industrial output value has experienced an astounding increase from CNY 22.1 trillion to CNY 156.57 trillion with a staggering growth rate reaching up to an astonishing figure of 608%. Therefore, solely evaluating the synergistic effect between air pollutants and carbon emissions based on total emission would be irrational and incomplete in nature. Recognizing this limitation, our study introduces the concept of an industrial output value in synergy index calculations while analyzing the interplay between $CO_2$ and air pollutants through changes in their emission intensity.

2.1.1. Synergy Index

Referring to the research of Mao [45] and Niu [46], this study introduces a quantitative synergy index to effectively characterize the synergistic effects of $CO_2$ and major air pollutants, namely $SO_2$, $NO_X$, and particulate matter. The specific calculations are presented in Equations (1)–(3).

$$S_{i,y} = \frac{\Delta CE_{i,y}/CE_{i,2005}}{\Delta PE_{i,y}/PE_{i,2005}} \tag{1}$$

$$CE_{i,y} = C_{i,y}/G_{i,y} \tag{2}$$

$$PE_{i,y} = P_{i,y}/G_{i,y} \tag{3}$$

where $S_{i,y}$ denotes the synergy index of industry $i$ in year $y$; $C_{i,y}$, $P_{i,y}$, and $G_{i,y}$ denote the carbon emissions, pollution emission, and industrial output of industry $i$ in year $y$, respectively. $CE_{i,y}$ and $PE_{i,y}$ represent the emission intensity of $CO_2$ and air pollutants in industry $i$ in year $y$. $\Delta CE_{i,y}$ is the change of $CO_2$ emission intensity of industry $i$

in year $y$ compared to the base year of 2005, and $\Delta PE_{i,y}$ is the change of air pollutant emission intensity.

The positive, negative and size of the synergistic effect index $S$ can reflect "whether there is synergistic effect" and "the degree of synergistic effect" between $CO_2$ and air pollutants in a certain industry. Figure 1 shows a schematic diagram of the synergy coordinate system. When the synergistic index falls within the range of points 1 and 2, a synergistic effect between $CO_2$ and air pollutants is observed, with closer y = x proximity indicating a stronger synergistic effect. Conversely, when the synergy index lies within the region of points 3 and 6, no synergistic relationship exists between $CO_2$ and air pollutants. Similarly, when the synergy index corresponds to points 4 and 5, it signifies an absence of synergistic emission reduction between $CO_2$ and air pollutants. Given the reality of China's industrial sector, the synergy index always falls in the region of points 1 and 2. When the index falls within point 1, it indicates a higher rate of reduction in air pollutant emissions compared to carbon emissions. Conversely, when the cooperation index falls within point 2, it suggests a faster reduction in carbon emissions relative to air pollutant emissions. To visualize the synergistic relationship between $CO_2$ and air pollutants, it is essential to standardize the calculated synergistic index. In cases where the synergy index exceeds 1, its reciprocal is computed. A higher value after normalization indicates a stronger synergistic effect between $CO_2$ and air pollutants.

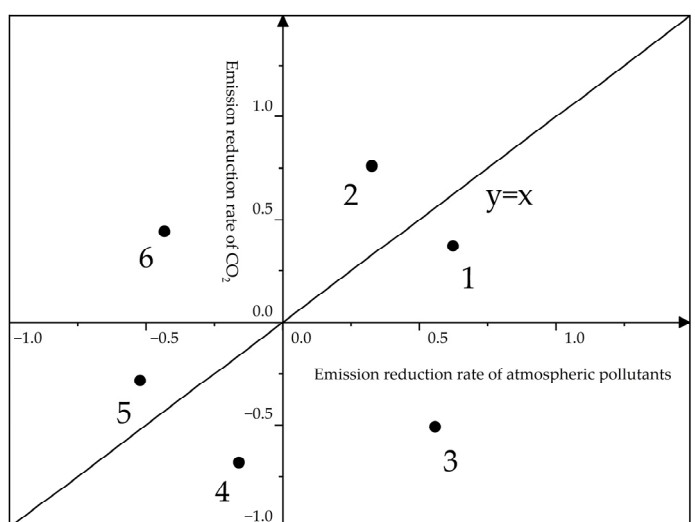

**Figure 1.** Schematic diagram of the synergy coordinate system.

### 2.1.2. $CO_2$ Emissions Calculation

The study employs the IPCC method to calculates $CO_2$ emissions of the industrial sector, as illustrated in Equation (4):

$$C_i = \sum_{j=1}^{27} E_{ij} \times NCV_j \times CCF_j \times COF_j \times \frac{44}{12} \tag{4}$$

where $C_i$ is the $CO_2$ emission of industry $i$; $j$ is the type of energy; $E_{ij}$ is the amount of energy $j$ consumed by industry $i$; $NCV_j$ is the average calorific value of energy $j$; $CCF_j$ is the carbon content per unit of calorific value of energy $j$; $COF_j$ is the oxidation rate of energy $j$.

### 2.1.3. Pollutant Emissions Calculation

The air pollutants examined in this study primarily consist of $SO_2$, $NO_X$, and particulate matter. To comprehensively assess the synergistic impact between $CO_2$ and multiple air pollutants in China's industrial sector, we employed the local air pollution equiva-

lent conversion factor method to standardize various air pollutants, as depicted using Equation (5):

$$E_{LAP} = \alpha E_{SO2} + \beta E_{NOX} + \gamma E_{PM} \tag{5}$$

where $E_{LAP}$ is the equivalent of air pollution; $E_{SO2}$, $E_{NOX}$, and $E_{PM}$ are the emissions of $SO_2$, $NO_X$ and particulate matter. $\alpha$, $\beta$, $\gamma$ are the equivalence coefficients of $SO_2$, $NO_X$, and particulate matter, respectively. Referring to the Law of the People's Republic of China on the Environmental Protection Tax, $\alpha$, $\beta$, $\gamma$ are the given values of 0.95, 0.95, and 2.18.

### 2.2. Data

Emissions of $SO_2$, $NO_X$, and particulate matter from the industrial sector used in this study were obtained from Table 4-3 of the China Statistical Yearbook On Environment [47]. The output value of the industrial sector was obtained from Tables 2-1 of the China Industrial Statistical Yearbook [48]. The amount of energy used was obtained from Tables 4-2 of the China Energy Statistics Yearbook [49]. The calorific value, carbon content, and oxidation-rate indicators used to calculate $CO_2$ emissions were from the Guidelines for the Preparation of Provincial Greenhouse Gas Inventories issued by the National Development and Reform Commission.

## 3. Results

### 3.1. The Collective Synergistic Effects of the Industry

The collective synergistic effects of the industry and the changes in pollution and carbon emission intensities during 2005–2021 are illustrated in Figure 2. In the period 2005–2021, there was a consistent decrease in the emission intensity of air pollutants, with levels dropping from 3.427 kg per 1000 RMB in 2005 to 0.074 kg per 1000 RMB. The emission intensity of $CO_2$ has exhibited an overall decline, decreasing from 156.108 kg per 1000 RMB to 50.262 kg per 1000 RMB. As a result, the obtained synergy index, which is greater than zero, indicates a significant synergistic effect between $CO_2$ and air pollutants within China's industrial sector as a whole.

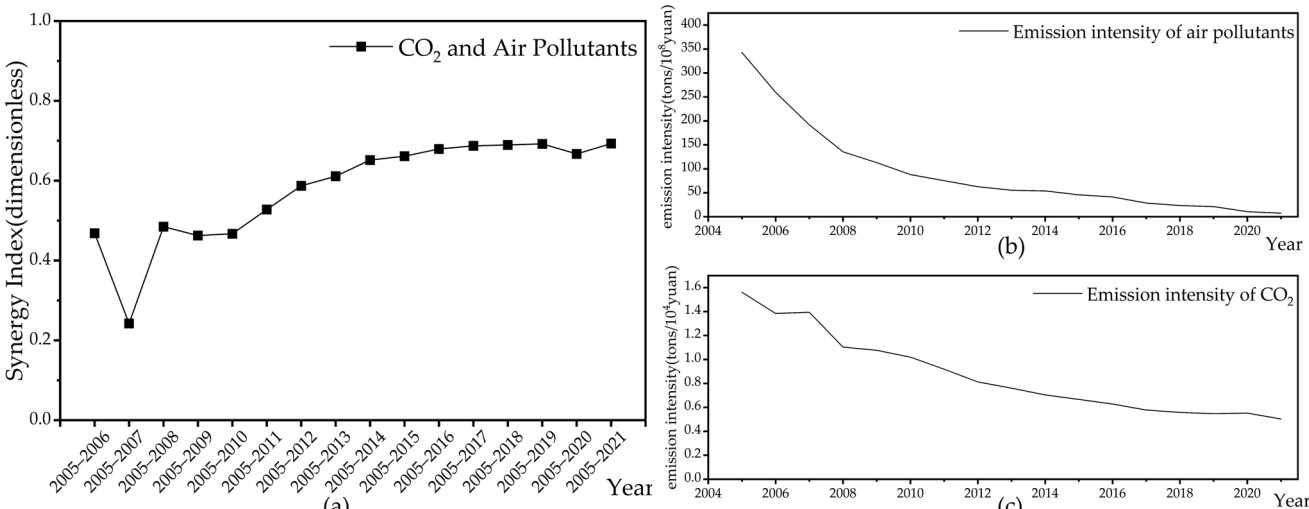

**Figure 2.** The synergy index and the changes in pollution and carbon emission intensities in 2005–2021. (**a**) Synergy index between $CO_2$ and air pollutants; (**b**) air pollutant intensity; (**c**) $CO_2$ emission intensity.

The synergistic relationship between $CO_2$ and air pollutants has undergone several stages. In the initial phase (2005–2009), the synergy index was unstable. It decreased due to the increase in $CO_2$ emission intensity in 2007, but increased when the emission intensity continued to decrease. The second phase (2009–2019) marked a significant shift, as the Chinese government first proposed at the Copenhagen Climate Conference in 2009 to reduce $CO_2$ emissions per unit of GDP by 40–45 percent compared to 2005 levels. This led to increased attention towards environmental hazards associated with greenhouse

gases, resulting in an increasing synergistic effect. Notably, regulatory measures such as the *"Cleaner Production Promotion Law"*, *"Energy Conservation and Emission Reduction Comprehensive Program"*, and others gradually transformed China's pollution management model from end-to-end management to source management and whole-process control. Additionally, during this period, the 12th Five-Year Plan for Greenhouse Gas Emission Work Program set a clear policy objective of reducing carbon emission intensity by 17% within five years. During the third phase (2010–2015), under strengthened source management and implementation of special measures, there was further enhancement of the synergistic effects between $CO_2$ and air pollutants during China's 13th Five-Year Plan period. Lastly, during the period of 2019–2021, while COVID-19 induced a decline in greenhouse gas emissions due to lockdown measures, the operation of fossil fuel-burning power plants and other essential industries persisted, resulting in an upsurge in air pollutant emissions such as $SO_2$. Consequently, the interplay between $CO_2$ and air pollutants exhibited a weakened synergy throughout 2019–2020. This synergy is anticipated to be reinforced with more incentive policies of pollution control and emission reduction.

### 3.2. Synergistic Effects between $CO_2$ and Different Pollutions

The synergistic indices between $CO_2$ and $SO_2$, $NO_X$, and particulate in 2005–2021 are shown in Figure 3. The average synergy indices between $CO_2$ and $SO_2$, $NO_X$, and particulate matter are 0.61, 0.58, and 0.57, respectively. The synergistic index between $CO_2$ and $SO_2$ is predominantly concentrated within the range of 0.55 to 0.68, while the index between $CO_2$ and particulate matter exhibits greater dispersion. These findings indicate that the synergistic effect between $CO_2$ and $SO_2$ as well as $NO_X$ in the industrial sector is marginally stronger compared to that with particulate matter.

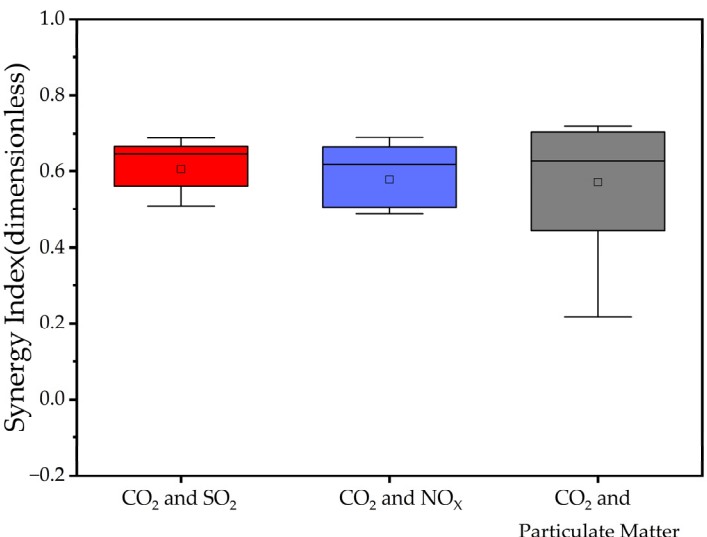

**Figure 3.** The synergy index between $CO_2$ and $SO_2$, $NO_X$, and particulate matter.

The high synergy index between $SO_2$ and $CO_2$ can be attributed to the high consistency in their emission sources. Coal dominates the energy consumption structure, serving as the primary source of both $CO_2$ and $SO_2$ emissions. In contrast, $NO_X$ sources are more diverse, with emissions arising from various energy consumption activities such as natural gas and oil usage. However, due to the relatively low proportion of carbon emissions originating from natural gas and oil within the industrial sector, the synergy index between $CO_2$ and $NO_X$ is slightly lower. Particulate matter originates not only from energy combustion but also contains a significant amount of industrial dust, resulting in a weaker synergistic relationship between particulate matter and $CO_2$ emissions.

### 3.3. Synergistic Effects between $CO_2$ from Different Energy Types and Pollutions

The synergistic index between $CO_2$ emissions from different energy sources and pollutants is illustrated in Figure 4. According to the emission sources, carbon emissions are categorized into five groups: coal, oil, natural gas, heat, and power. The latter two types represent indirect sources.

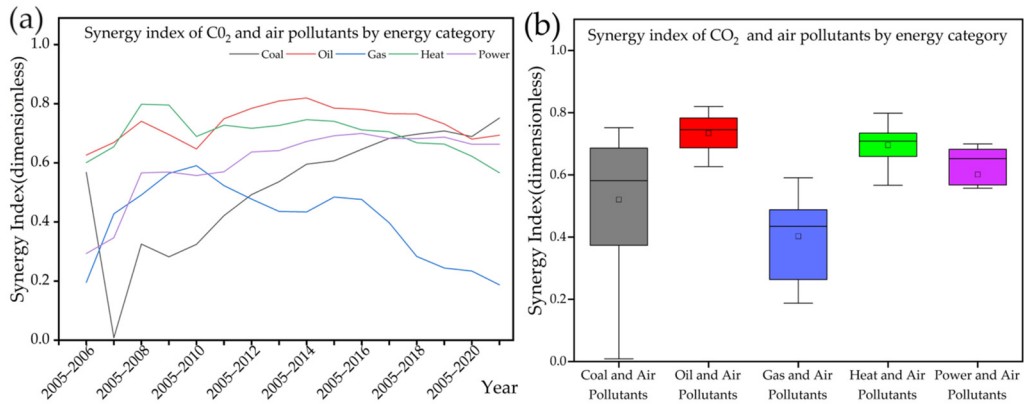

**Figure 4.** Synergy index between $CO_2$ emissions from different energy sources and pollutants, 2005–2021. (**a**) the change of synergy index in 2005–2021; (**b**) distribution of synergy index.

The trend in the synergies index from 2005 to 2021 is shown in Figure 4a. Overall, the synergistic index between coal-related $CO_2$ and pollutants continues to increase, while the synergistic index between natural-gas-related $CO_2$ and pollutants shows a trend of first increasing and then decreasing. The lower synergistic index between $CO_2$ and the pollutant index associated with coal in 2007 was due to a significant increase in coal consumption and an increase in coal intensity per unit of industrial output. Before China's air pollution control campaign began in 2013, companies mainly reduced air pollutant emissions through end-stage measures such as desulfurization and denitrification, which significantly reduced pollutant emissions but did not reduce carbon dioxide emissions. With the gradual implementation of de-coal policies, the synergistic relationship between pollutants and $CO_2$ emissions began to rise rapidly. After 2010, the synergy index between gas-related carbon emissions and pollutants continued to decrease, primarily because the carbon emissions in this study came from natural gas, while the pollutants came from industry. As the proportion of gas consumption increases, gas-related carbon emissions continue to increase while overall air pollutants decrease, resulting in a gradual weakening of the synergistic relationship between gas-related carbon emissions and air pollutants.

The general picture of the synergies index for the years 2005–2021 is shown in Figure 4b. The synergistic index between carbon emissions and pollutants associated with oil, heat, and power are relatively stable overall, while those associated with coal and natural gas vary considerably. Total $CO_2$ emissions related to oil and heat are low, and the trend of decreasing $CO_2$ intensity is generally consistent with the trend of decreasing pollutant intensity. The synergy between electricity-related carbon emissions and pollutants is generally high and has continued to increase since 2005. The main reason is that the proportion of non-fossil energy generation continues to increase and carbon emissions per unit of electricity continue to decline, which drives continued reductions in electricity-related carbon intensity and reinforces the synergistic relationship between electricity carbon emissions and pollutants.

### 3.4. Synergistic Effects of Different Industry

The synergistic effects of $CO_2$ emissions and air pollutants in different sectors are shown in Figure 5. Overall, there is a clear synergy effect between carbon emissions and air pollutants across industries. Among them, the synergistic index of the mineral mining industry (3. Mining and Processing of Ferrous Metal Ores; 4. Mining and Processing of

Non-Ferrous Metal Ores), daily necessities processing industry (15. Processing of Timber, Manufacture of Wood, Bamboo, Rattan, Palm, and Straw Products; 24. Manufacture of Rubber and Plastics Products), machinery manufacturing industry (30. Manufacture of Special Purpose Machinery; 32. Manufacture of Railway, Ship, Aerospace and Other Transport Equipment) and other industries is relatively high and the inter-annual change is small. The synergistic index of the mining service industry (6. Professional and Support Activities for Mining; 7. Mining of Other Ores), furniture manufacturing industry (16. Manufacture of Furniture), metal products, machinery and equipment repair industry (38. Repair Service of Metal Products, Machinery and Equipment), and other industries varied greatly from year to year, and there was no coordination in some years. This may be due to the fact that the products of these industries are updated at a fast pace and the sources of raw materials are highly dependent on other industries. With the change of people's consumption concepts, outdated products are not recognized by the public, leading to overcapacity. The production process mostly adopts end-of-pipe treatment to reduce the emission of air pollutants, leading to a large change in the synergy index.

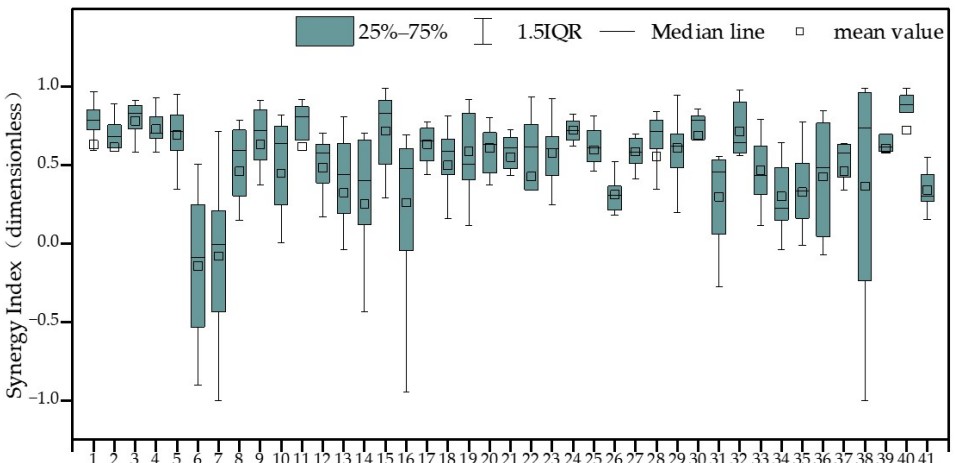

**Figure 5.** Synergy index between $CO_2$ and air pollutants for different industries. Note: The industries represented by the numbers in the chart are 1. Mining and Washing of Coal; 2. Extraction of Petroleum and Natural Gas; 3. Mining and Processing of Ferrous Metal Ores; 4. Mining and Processing of Non-Ferrous Metal Ores; 5. Mining and Processing of Nonmetal Ores; 6. Professional and Support Activities for Mining; 7. Mining of Other Ores; 8. Processing of Food from Agricultural Products; 9. Manufacture of Foods; 10. Manufacture of Liquor, Beverages and Refined Tea; 11. Manufacture of Tobacco; 12. Manufacture of Textile; 13. Manufacture of Textile, Wearing Apparel and Accessories; 14. Manufacture of Leather, Fur, Feather and Related Products and Footwear; 15. Processing of Timber, Manufacture of Wood, Bamboo, Rattan, Palm, and Straw Products; 16. Manufacture of Furniture; 17. Manufacture of Paper and Paper Products; 18. Printing and Reproduction of Recording Media; 19. Manufacture of Articles for Culture, Education, Arts and Crafts, Sport and Entertainment Activities; 20. Processing of Petroleum, Coal, and Other Fuels; 21. Manufacture of Raw Chemical Materials and Chemical Products; 22. Manufacture of Medicines; 23. Manufacture of Chemical Fibers; 24. Manufacture of Rubber and Plastics Products; 25. Manufacture of Non-metallic Mineral Products; 26. Smelting and Pressing of Ferrous Metals; 27. Smelting and Pressing of Non-ferrous Metals; 28. Manufacture of Metal Products; 29. Manufacture of General Purpose Machinery; 30. Manufacture of Special Purpose Machinery; 31. Manufacture of Automobiles; 32. Manufacture of Railway, Ship, Aerospace, and Other Transport Equipment; 33. Manufacture of Electrical Machinery and Apparatus; 34. Manufacture of Computers, Communication, and Other Electronic Equipment; 35. Manufacture of Measuring Instruments and Machinery; 36. Other Manufacture; 37. Utilization of Waste Resources; 38. Repair Service of Metal Products, Machinery, and Equipment; 39. Production and Supply of Electric Power and Heat Power; 40. Production and Supply of Gas; 41. Production and Supply of Water.

In 2021, the Ministry of Ecology and Environment issued a guideline on strengthening the prevention and control of the use of ecological environment as the source of high-energy-consumption and high-emission construction projects. In which, six industry categories such as coal power, petrochemicals, chemicals, iron and steel, non-ferrous metal smelting, and building materials are categorized as high-energy-consumption and high-emission projects, in order to better understand the overall situation of the synergy index of different industries. With reference to the content of the guidance, we define the following industries as high-energy-consumption and high-emission industries. Such as 1. Mining and Washing of Coal; 2. Extraction of Petroleum and Natural Gas; 3. Mining and Processing of Ferrous Metal Ores; 4. Mining and Processing of Non-Ferrous Metal Ores; 5. Mining and Processing of Nonmetal Ores; 6. Professional and Support Activities for Mining; 7. Mining of Other Ores; 17. Manufacture of Paper and Paper Products; 20. Processing of Petroleum, Coal and Other Fuels; 21. Manufacture of Raw Chemical Materials and Chemical Products; 24. Manufacture of Rubber and Plastics Products; 25. Manufacture of Non-metallic Mineral Products; 26. Smelting and Pressing of Ferrous Metals; 27. Smelting and Pressing of Non-ferrous Metals; 39. Production and Supply of Electric Power and Heat Power.

From Table 1, it can be seen that the high-energy-consumption and high-emission industries have a high proportion of $CO_2$, $SO_2$, $NO_X$, and particulate matter emissions. It shows that the pollution-reduction and carbon-reduction synergy of high-energy-consumption and high-emission industries are of great significance to the pollution-reduction and carbon-reduction work of the whole industry. The synergy index of the two industries is shown in Figure 6. Overall, the average synergistic index for $CO_2$ and air pollutants in high-polluting and high-emission industries was 0.56, and the average synergistic index of other industry was 0.67. Industries that are highly polluting and produce high emissions have weaker synergies between $CO_2$ and air pollutants than other industries. The low synergy index for high-polluting and high-emission industries is due to their complex process routes, high dependence on coal and other fossil energy sources, and reliance on end treatment for pollutant-reduction measures. In contrast, energy use facilitated in other industries is mainly for general facilities such as boilers, pumps, and motors, and the reduction of pollutants and carbon dioxide emissions in these industries is more dependent on energy substitution and energy-saving technologies, so that pollutants and carbon dioxide emissions have a higher degree of synergy.

**Table 1.** Percentage of $CO_2$, $SO_2$, $NO_X$, and particulate matter emissions from high-energy-consuming and high-emission industries.

| Year | $CO_2$ | $SO_2$ | $NO_X$ | PM |
|---|---|---|---|---|
| 2005 | 0.83 | 0.94 | 0.91 | 0.94 |
| 2006 | 0.83 | 0.94 | 0.91 | 0.94 |
| 2007 | 0.82 | 0.94 | 0.89 | 0.94 |
| 2008 | 0.82 | 0.94 | 0.89 | 0.93 |
| 2009 | 0.83 | 0.93 | 0.89 | 0.93 |
| 2010 | 0.83 | 0.93 | 0.89 | 0.92 |
| 2011 | 0.84 | 0.93 | 0.97 | 0.89 |
| 2012 | 0.84 | 0.92 | 0.97 | 0.90 |
| 2013 | 0.85 | 0.92 | 0.97 | 0.91 |
| 2014 | 0.85 | 0.92 | 0.96 | 0.93 |
| 2015 | 0.85 | 0.91 | 0.96 | 0.92 |
| 2016 | 0.85 | 0.90 | 0.93 | 0.85 |
| 2017 | 0.85 | 0.95 | 0.95 | 0.88 |
| 2018 | 0.85 | 0.95 | 0.95 | 0.88 |
| 2019 | 0.85 | 0.94 | 0.94 | 0.87 |
| 2020 | 0.86 | 0.94 | 0.96 | 0.95 |
| 2021 | 0.84 | 0.96 | 0.96 | 0.96 |
| Average | 0.84 | 0.93 | 0.94 | 0.92 |

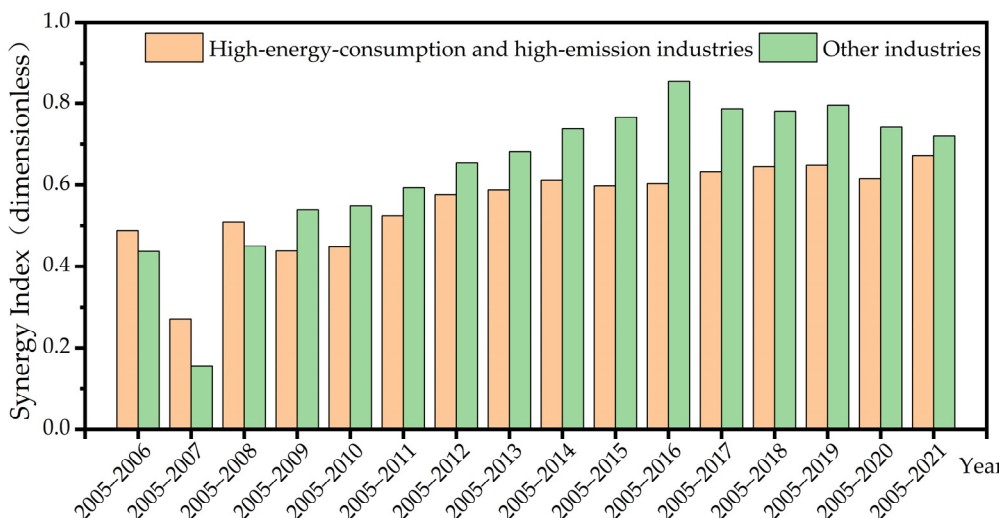

**Figure 6.** The synergy index between $CO_2$ and air pollutants for high-energy-consumption and high-emission industries and other industries, 2005–2021.

## 4. Policy Implications

### 4.1. The Focus of Synergistic Measures Should Be Directed towards Source Reduction

The energy structure of China's industrial sector is dominated by fossil fuels such as coal. Fossil energy is an important source of air pollution and $CO_2$ emissions, and controlling fossil energy consumption is a core measure to achieve the synergistic effect of reducing pollution and carbon. From an energy-type perspective, there is a natural synergy between $CO_2$ and $SO_2$, particulate matter for coal; $CO_2$ and $NO_X$, particulate matter for oil; and $CO_2$ and $NO_X$ for natural gas. For a long time, China's industrial sector has paid more attention to reducing pollutants and setting targets for overall reductions. The emphasis on $CO_2$ reduction is relatively weak, with only emission intensity targets set in policy. This has led the industrial sector to take more drastic measures to meet environmental targets [50]. With the continuous reduction of total pollutant emissions, it is difficult to achieve further reduction of pollutants only by relying on end measures, and we must rely on source measures such as energy conservation and energy structure adjustment to achieve environmental goals, which will have a significant $CO_2$-emission-reduction effect [51]. At the same time, China's environmental policy has gradually shifted from pollutant reduction to carbon peak and carbon neutrality, and the industrial sector must also rely on source measures to meet its carbon peak and carbon neutrality targets.

### 4.2. The Industries Characterized by Being Highly Polluting and Having High Emissions Play a Pivotal Role in Fostering Synergistic Effects

The high-polluting and high-emission industries are the mainstay of energy consumption and play a leading role in industrial pollutants and $CO_2$ emissions. However, due to the high-polluting and high-emission industrial process routes, it is difficult for these industries to implement energy substitution, and traditional measures are often dominated by end treatment, resulting in a low degree of synergy between $CO_2$ and pollutants. To promote synergies across society for reducing pollution and carbon in the future, the government must focus on industries with high energy consumption and emissions. There are two ways to promote collaborative emission reduction in industries that are highly polluting and have high emissions: first, reform traditional process routes through scientific and technological innovation [52], such as short-process steelmaking and hydrogen steelmaking, and promote energy substitution in industries with high energy consumption and high emissions. Second, we must eliminate backward production capacity [53]. Many of China's high-polluting and high-emission industries have significant overcapacity. The elimination of backward production capacity will not only help the healthy development

of the industry, but also reduce pollutants and $CO_2$ emissions at the source and achieve coordinated pollution reduction and carbon reduction.

*4.3. The Dual Management of Air Quality and Carbon Emissions Peaks Facilitates the Promotion of Synergistic Effects*

China's environmental policy has long been characterized by "put more emphasis on pollution reduction than carbon reduction," and the decline in carbon intensity in China's industrial sector before 2020 is largely driven by air quality improvement measures. With carbon peak and carbon neutrality becoming national strategies, the priority of carbon-reduction measures in environmental policy is rapidly increasing, and future coordination of carbon reduction and pollution reduction will depend more on the implementation of carbon-reduction policies. To promote the synergistic effect of pollution reduction and carbon reduction, it is necessary to coordinate air quality improvement with the goal of carbon peaking and carbon neutrality, and to implement air quality compliance and carbon peaking management [54]. Currently, China's air quality is managed by the ecological environment department; the carbon peak is managed by the National Development and Reform Commission; industrial production is managed by the industry and information technology departments; and the industry management of carbon reduction still needs to be reformed [55]. We propose providing priority to the coordinated monitoring and statistics of pollutants and carbon emissions in the industrial sector to lay the data foundations for pollution-reduction and carbon-reduction actions. Based on this, departments should work together to develop pollution and $CO_2$-reduction measures, reduce the costs of reducing pollution and $CO_2$, and promote the achievement of air quality and $CO_2$ peak targets.

## 5. Research Limitations

The data for the calculation of the synergy index came from the statistical yearbook, and the arithmetic average interpolation method is used for the missing data, which has certain uncertainties. The $CO_2$ emissions in the calculation of synergy index came from the amount of energy used, and the change of synergy index reflects the impact of energy structure adjustment on the synergy effect to a certain extent. The synergy index is the result of qualitative analysis, and it is a limitation of this study that it cannot reflect the influence of the adjustment of the industrial structure of specific industries on the synergy effect. The influence of industry-specific industrial structure adjustment and energy structure adjustment on the synergy effect is a future research direction.

## 6. Conclusions

Based on carbon emissions and pollutants such as $SO_2$, $NO_X$, and particulate matter from 2005 to 2021 in the Chinese industrial sector, we construct a synergistic index for pollution reduction and carbon reduction in the industrial sector. On this basis, the synergistic effect between $CO_2$ and pollutants in the industrial sector is analyzed, and the following conclusions are drawn: (1) From 2005 to 2021, there is a clear synergistic effect between $CO_2$ and air pollutants in China's industrial sector, and this synergistic effect is constantly increasing. (2) In terms of pollutant types, the synergistic effect between $CO_2$ and $SO_2$ is the strongest, while the synergistic effect between $CO_2$ and particulate matter is relatively weak, mainly because $SO_2$ mainly comes from energy combustion, while particulate matter comes from both energy combustion and the industrial production process. (3) From the perspective of energy type, the synergistic effect between coal-related carbon emissions and air pollutants is gradually enhanced, while the synergistic effect between gas-related carbon emissions and pollutants is decreasing. (4) The synergistic effect between $CO_2$ and air pollutants in high-polluting and high-emission industries is weaker than that in other industries, mainly because of the limited emission-reduction measures at the source of high-polluting and high-emission industries. To promote pollution reduction and carbon reduction in the industrial sector, we propose three potential policy recommendations: First, the focus of synergistic measures should be on source reduction. The second is to

make high-polluting and high-emission industries the focus of pollution reduction and carbon reduction. Third is the harmonized management of air quality standards and carbon peaking should be promoted.

**Author Contributions:** Conceptualization, H.L.; methodology, M.X.; formal analysis, M.X.; resources, H.L. and X.D.; writing—original draft preparation, M.X.; writing—review and editing, M.X. and H.L.; supervision, H.L. and X.D.; funding acquisition, H.L. All authors have read and agreed to the published version of the manuscript.

**Funding:** This research was funded by the Project of Construction and Support for high-level Innovative Teams of Beijing Municipal Institutions (BPHR20220108).

**Institutional Review Board Statement:** Not applicable.

**Informed Consent Statement:** Not applicable.

**Data Availability Statement:** Data are contained within the article.

**Conflicts of Interest:** The authors declare no conflicts of interest.

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
