# Peer review of "Measuring the Synergistic Effect of Pollution and Carbon Reduction in China’s Industrial Sector"

_sustainability, doi:10.3390/su16031048_

Round 1

Reviewer 1 Report

Comments and Suggestions for Authors

This paper employs a synergy index method to assess the pollution and carbon reduction in China's industrial sector. The following comments should be addressed:

1.       Line 21: Please specify the policy implications.

2.       Line 85-91: This sentence is too long; please break it into shorter ones.

3.       Introduction: Please clearly indicate what is the innovation point of your paper in the context of previous studies and the broader scientific literature.

4.       Line 103-108: This seems to be the three gaps in the field, but the results and conclusion of the paper do not clearly address the three gaps.

5.       Methods: Please provide the sources, characteristics, and uncertainties for the 2005-2021 raw data used in calculations.

6.       Methods: Could the normalizing using industrial output value distort the results? What is the rationale behind this approach? Any references?

7.       Results: There are many atmospheric pollutants in industrial sectors. Why choose only SO2, NOX, and particulate matter?

8.       Results: Figure 6: What are the criteria for determining high-energy-consuming and high-emission industries?

9.       Discussion: The discussion contains too much speculates; more literatures are needed to strengthen the discussions.

10.     Conclusion: The assertion that source control measures are limited in high-energy-consuming and high-emission industries seems illogical. Please clarify.

Comments on the Quality of English Language

Please break long sentences into shorter ones.

Author Response

Thank you very much for your modifications. We have revised the article by listening to your suggestions. Please see the attached document for details.

Reviewer 2 Report

Comments and Suggestions for Authors

1.Please define the "high-polluting and high-emission industries".

2.Please present the ratio of emissions (SO2, NOx, PM and CO2) from “high-polluting and high-emission industries” to those from total emissions of industrial sector. It’s important to understand the trend of synergy index.

3.Fig.5 provides valuable information. Several industries, such as 6,7,16,31 and 38, present great variation. Does the synergy index fit in these industries?

4.It’s better to provide the information of structural adjustment of the industrial scale and energy during these years. It’s important to assess the applicability of synergy index as universal index.

Author Response

(The authors gave the same response as above.)

Reviewer 3 Report

Comments and Suggestions for Authors

Dear authors

Lines 119-122 can be omitted because, in the further exposition, the parts of the exhibited work are listed individually.

  Part 2.1. should be moved to Introduction.

  Part 2.2. transfer instead of part 2.1. and list the references or more, if any, from which they are taken, with a clear indication of where the source of the processed data comes from.

The specified formulas need to be centered.

Units of measure should be indicated following the SI system (line 208).

The sentence in line 240 needs to be reformulated so that it does not start with Figure 3.

From line 268 onwards, where there is no text edit (Justify)

In lines 297-304, it should be indicated that the numbers shown are the type of industry to which the research refers.

Notes to figure 6, if it is possible to explain in the text.

Check the references and add foreign codes, for example, references 15 and 16, respectively 29.

Author Response

(The authors gave the same response as above.)

Round 2

Reviewer 1 Report

Comments and Suggestions for Authors

The authors have adequately addressed the comments, and I have no further comments.

Author Response

Dear Reviewer

Thank you for commenting and recognizing our articles!

Best Wishes

Minglong Xu